# Has the Regulatory Compliance Burden Reduced Competitiveness of the U.S. Tilapia Industry?

**Carole R. Engle [1],\*, Jonathan van Senten [2], Charles Clark [2] and Noah Boldt [2]**

[1] Engle-Stone Aquatic$ LLC, Strasburg, VA 22657, USA
[2] VA Seafood AREC, Virginia Polytechnic Institute and State University, Blacksburg, VA 24061, USA
\* Correspondence: cengle8523@gmail.com

**Abstract:** Emerging research on aquaculture governance has pointed to the conundrum of negative global environmental effects from economic incentives for aquaculture production to shift from more highly regulated to less regulated countries. This study has focused on examining whether regulatory costs on U.S. tilapia farms may have contributed to their contraction in contrast to the growth of global tilapia production that contributes to the volume of seafood imports into the U.S. A national survey (coverage rate = 75% of tilapia sold; response rate = 18%) found that on-farm regulatory costs accounted for 15% of total production costs on U.S. tilapia farms, the fifth-highest cost of production. The total direct regulatory costs nationally were $4.4 million, averaging $137,611/farm. Most problematic were regulations of effluent discharge, predatory bird control, international export, and water and energy policies. Manpower costs for monitoring and reporting were the greatest cost of regulatory compliance. The lost sales revenue resulting from regulations was $32 million a year, or 82% of total annual sales, indicating that the regulatory framework has constrained the growth of U.S. tilapia farming. The smallest tilapia farms had the greatest regulatory cost per kg. This study provides evidence that regulatory costs, along with other challenges related to live fish markets, have contributed to the decline in U.S. tilapia production. Increased competitiveness of the U.S. tilapia industry will require a combination of: (1) improved regulatory efficiency that reduces on-farm cost burdens without reducing societal benefits; (2) research and on-farm extension assistance to evaluate new tilapia fillet equipment; and (3) research on changing consumer preferences to provide guidance on effective strategies to penetrate the large U.S. fillet market.

**Keywords:** tilapia; U.S. tilapia farming; regulatory costs; aquaculture economics; regulations

**Key Contribution:** The regulatory compliance burden was the fifth-greatest cost on U.S. tilapia farms and, combined with the loss of sales revenue resuting from the regulatory framework, has likely contributed to contraction in the U.S. The disparity in regulatory enforcement and control between developed countries like the U.S. and EU with developing countries has likely created perverse economic incentives for aquaculture production to shift to countries with less rigorous environmental management control.



## 1. Introduction

The continued growth of the global human population has led to serious challenges of meeting global food production needs while reducing the often accompanying environmental and social costs [1]. Effective governance has become ever more critical as the demand for food, but also for improved environmental and social quality, has increased. Safeguarding the environment and individuals within society requires the promulgation of laws, regulations, and rules that are accompanied by effective enforcement [2]. The absence of effective governance leads to negative externalities that include pollution and environmental contamination, as well as unsafe and unhealthy workplace conditions [3,4].

Research efforts in the 1960s and 1970s developed into new disciplines such as environmental science [5] and environmental economics [6] that focused on the reduction of negative externalities to the environment and society that resulted from growth in the human population and its associated economic development. Attention to the environmental problems that resulted from unregulated, negative externalities led to the promulgation of comprehensive environmental laws in many countries. Those countries that successfully implemented effective laws and regulatory structures have benefitted over the years in various ways from the resulting improvements in environmental quality, workplace safety, and public health [7].

Aquaculture entered a phase of rapid development of new, more efficient, and intensive production during the 1970s and 1980s [8], the same period during which a host of new governance initiatives to improve environmental and social quality were enacted in the U.S. and EU. By the late 1970s, research had begun to emerge that raised questions as to whether the U.S. regulatory framework with respect to aquaculture had created unintended adverse effects on the sustainability of aquaculture farm businesses [9,10]. Studies emerged in the 2000s that broadly examined negative regulatory effects on U.S. agriculture [11–14]. By 2012, a U.S. Executive Order had called for reduction of the "billions of dollars in regulatory costs and tens of millions of hours in annual paperwork burdens" [15].

The early studies on the governance of aquaculture laid the foundation for a new research discipline that focused on the more specific economic effects of the regulatory framework on aquaculture. Two lines of regulatory governance research emerged. In more developed countries such as the EU and the U.S., research focused on the degree to which aquaculture farm businesses may have faced inefficient, overlapping monitoring and compliance requirements that have reduced the competitiveness of U.S. aquaculture [16–20]. Personal ideologies of agency personnel were found to exacerbate regulatory costs and constraints on aquaculture businesses in the EU [21]. The prescriptive approaches frequently used were further shown to lack flexibility to adjust regulations to the rapid pace of technological advancement of aquaculture production practices [4], thereby negatively affecting the ability of businesses to adopt new and improved technologies [22]. New technologies in some cases, if allowed to be used, would have reduced the environmental footprint of aquaculture [23]. Domestic regulations have also reduced the productivity of aquaculture farms [24,25], thwarted future growth [17,26], and resulted in overly complex and redundant monitoring and reporting requirements that have led to excessive costs [27]. The studies that examined the broad trends in regulatory effects raised important questions about the effects of regulations on the economic sustainability of individual farms [19,28]. An analysis of farm-level data found that Norwegian salmon farms subjected to unusually high regulatory burdens were less efficient [29]. Similarly, farm-level inefficiencies on Chilean salmon farms have been attributed to regulatory burdens [30]. In the U.S., sector-specific surveys of the U.S. catfish, salmonid, Florida tropical, and Pacific Coast shellfish sectors found that the direct cost of regulations accounted for 8% to 29% of total production costs, making regulatory costs one of the top five costs of production [23,31–34]. Moreover, regulations were found to have curtailed expansion due to the loss of markets, the loss of production that prevented farms from achieving economies of scale, and other lost business opportunities. Total revenue lost summed to more than $360 million annually across the catfish, salmonid, Florida tropical, and Pacific Coast shellfish sectors [23,31–34]. Insights from these studies show that the cost of the required permits was a very minor cost; the major costs accrued from the time spent on record keeping and reporting as well as seeking out specifics of frequently changing and unannounced regulatory compliance requirements. For some sectors, the value of lost or foregone revenue far exceeded the direct regulatory costs. Moreover, in the U.S., the regulatory compliance requirements varied widely across states, both for farms located within a given state and for farms seeking to sell live fish into other states [31,32,34–36].

The second line of research that has developed related to the governance of aquaculture focused on the need for developing countries to implement "sound rule of law, low levels of corruption, and effective government services" [3]. Studies have shown that the lack

of adequate governance in aquaculture has led to declining environmental quality [37]. Negative environmental effects have been documented from poorly sited and operated shrimp farms [38], salination of farm land and drinking water [39], nutrient pollution from cage culture in the Philippines [40], and loss of mangrove areas [41]. From a food safety perspective, the unregulated use of antibiotics and chemicals in aquaculture [42–45] has led to health risks from residual levels of compounds of concern in the environment but also in food consumed by local residents [46,47]. The discharge of nutrients from aquaculture facilities in China accounted for 20% of total nutrient inputs into freshwater resources [48], which are widely used for drinking water by local residents [49].

These two lines of research (from developed and developing countries) intersect in the global seafood marketplace that has emerged since the 1970s [50,51], with important implications for global environmental and economic sustainability. For example, countries with less stringent and less effective environmental protection regulatory frameworks were found to be those where aquaculture has grown rapidly, whereas in the US and EU, strict environmental regulations have constrained its growth [18]. The shift in aquaculture production to countries without effective environmental protection and regulatory control systems likely results in a net negative effect on global environmental quality. This results from a perverse cost incentive to increase production in countries without the stringency and high regulatory cost, thereby leading to greater environmental degradation.

Given the magnitude of the challenge of feeding the global human population with healthy food, greater attention to the reform of aquaculture regulations is needed. In developed countries, identifying sources of inefficiency and the highest governance costs is a necessary step in the design and implementation of more efficient regulatory processes that support the economic sustainability of aquaculture. In many developing countries, the development of more effective governance and enforcement frameworks that approach the stringency of developed countries is essential to reducing negative environmental effects.

The key research question that has emerged from the knowledge base on regulatory governance for aquaculture in developed countries is to identify which regulations are most costly and most constraining for both well-established and newer, emerging sectors. Detailed information on farm-level regulatory costs and lost market revenue effects is an essential base from which to identify cost-efficient alternatives for monitoring, reporting, and permitting. This paper contributes to the knowledge base of the farm-level effects of regulations by examining their economic effects on U.S. tilapia farms.

Globally, Nile tilapia (*Oreochromis niloticus*) was the third-most important finfish by volume in 2018, with a global annual average production of 14% that has increased continuously from 2010 through 2018 [44]. Tilapia are farmed around the world on a wide range of production scales that vary from extensive to intensive production in open ponds, cages, outdoor tanks (both flow-through and recirculating), and indoors in flow-through tanks, traditional recirculating aquaculture (RAS), and biofloc systems [52,53].

Tilapia farming dates back more than 3500 years to Egypt. Production practices used worldwide through the 1950s were extensive, low intensity methods used by subsistence farmers [54]. Major technological advancements in the latter half of the 20th century led to effective means to control reproduction, improved feeds and genetic stocks, and innovative production systems that improved efficiencies and led to the rapid growth of tilapia production worldwide [55].

The U.S. is the largest seafood market in the world, consuming $102 billion of seafood products annually [56]. Of this, an estimated 79% is imported [57]. Tilapia is the fourth-most consumed seafood in the U.S., after shrimp, salmon, and tuna [58]. However, tilapia imports into the U.S. peaked in 2014 and declined by approximately 20% from 2014 to 2017 (the most recent trade data available) [59]. The majority of tilapia imports are frozen and primarily in fillet form. The decline of tilapia sales in the U.S. has been attributed to a combination of factors that include concerns over the quality of frozen imported product and tariffs on Chinese tilapia sold to the U.S., among others [60].

In the U.S., tilapia have been farmed since the early 1970s [61,62], and production has grown to become the fourth-largest sector of food fish production. Marketed first as aquarium fish that were imported from their native countries, tilapia was later introduced to natural waters in Florida and California. A commercial fishery developed for tilapia in Lake Okeechobee, Florida, in the 1970s, which prompted market development and greater interest in the potential for tilapia as a commercially farmed product. The earliest published marketing study on tilapia showed that U.S. consumers found tilapia to be an acceptable substitute for "perch" and other freshwater fish [63]. The U.S. Census of Aquaculture [64] shows that more than half (56%) of U.S. states have at least one tilapia farm (Figure 1), with farms spread across both coasts and in interior, northern, and southern states. Four important tilapia farm clusters (California, North Carolina, Idaho, and Florida) account for nearly half (46%) of reported tilapia sales [64], with a few large farms in other states. Nationally, there were 137 farms reported in 2018 with sales of $39.4 million (USD), making tilapia the fourth-largest food fish sector of U.S. aquaculture.

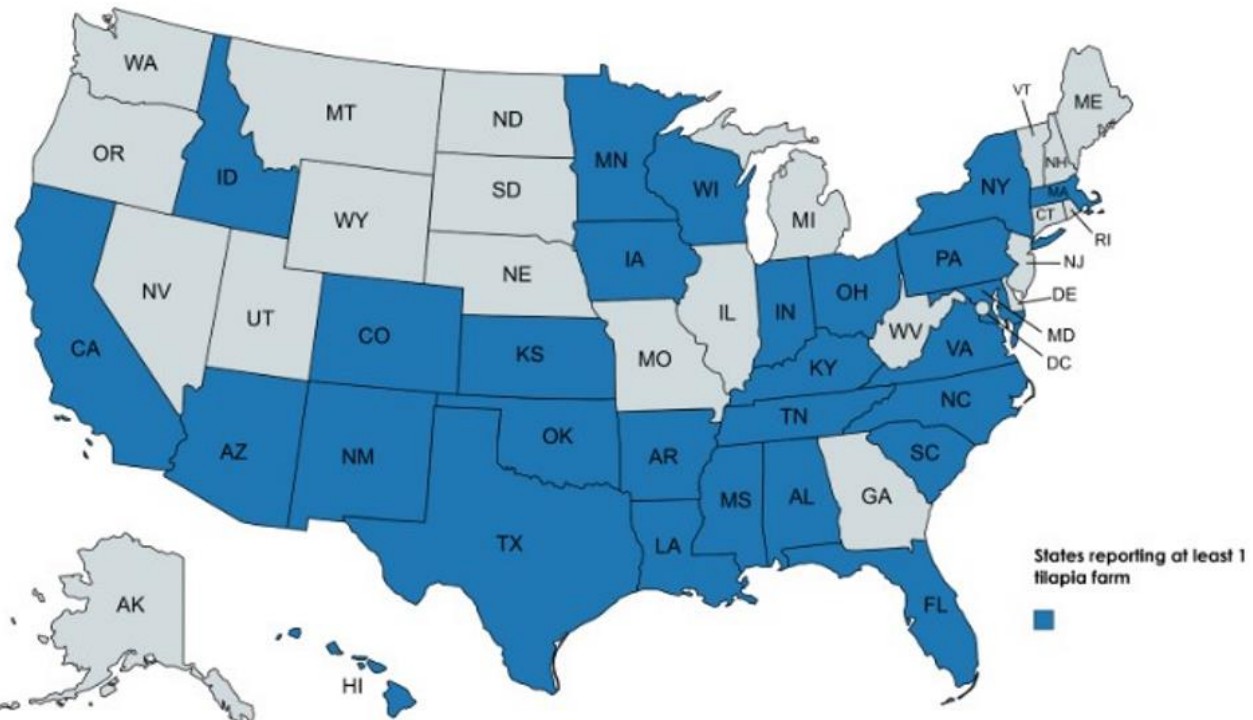

**Figure 1.** U.S. states reporting at least one tilapia farm in 2018. SOURCE: 2018 Census of Aquaculture (USDA-NASS 2019).

Tilapia sales and the number of tilapia farms peaked, however, in 2012 at $42.5 million in sales and 181 farms reported (Figure 2) [64–66]. Tilapia sales declined by 7% from 2012 to 2018, and the number of farms raising tilapia decreased by 24%, to less than the number of farms reported in the 2005 Census of Aquaculture. Over this same time period (2012 to 2018), the global annual production of tilapia increased by 35% [51]. Thus, there are important questions related to the competitiveness of U.S. tilapia production, given the recent declines in the U.S. that contrast sharply with the global increase in tilapia sales. Possible hypotheses for the contraction of U.S. tilapia production include: (1) greater production costs as a result of the regulatory framework; (2) substitution of other live fish in the markets targeted by U.S. tilapia producers; or (3) other economic factors.

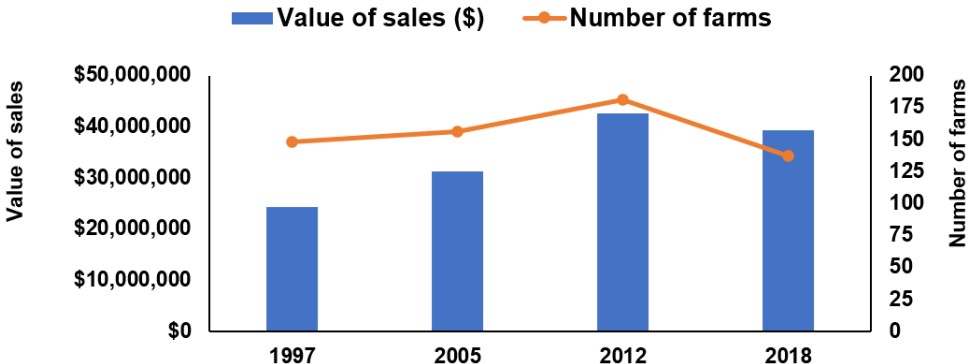

**Figure 2.** U.S. tilapia sales and number of farms, 1997 to 2018. SOURCE: USDA-NASS.

A key research motivation of this study was to examine the case of tilapia from the perspective of the effects of regulatory governance. Tilapia are raised in both developed and developing countries and are exported in large volumes to the U.S. and the EU. In the U.S., it is possible that the regulatory framework for tilapia farms may have contributed to its decline. If so, a shift of tilapia production to countries with less stringent regulations could have greater negative effects on the global environment than if the demand for tilapia were met by U.S. producers, who are held to high environmental standards. A secondary motivation for this study is that there is no information on the economics of U.S. tilapia farming, despite the many tilapia economic studies in other countries [67–71]. Moreover, the magnitude of regulatory costs on U.S. tilapia farms and the extent to which the regulatory framework may or may not have contributed to their contraction are unknown.

The overall goal of this study was to improve understanding of the challenges faced by U.S. tilapia farmers and to evaluate the extent to which regulatory compliance burdens may have contributed to the declining competitiveness of tilapia production in the U.S. Specific objectives included: (1) conducting a national survey of tilapia farms in the U.S.; (2) identifying key challenges for tilapia farms; (3) measuring the economic effects of regulatory compliance on U.S. tilapia farms; and (4) evaluating whether regulatory constraints have contributed to the contraction of the U.S. tilapia sector. This paper is expected to fill several important gaps in our understanding of the economics of aquaculture production by studying one of the major food fish sectors in U.S. aquaculture, for which little work has been done.

## 2. Methods

### 2.1. Study Design

The study was designed as a descriptive cross-sectional research survey. A descriptive research survey was chosen as the study design because no previous studies had been done previously on U.S. tilapia farms to identify key economic characteristics of tilapia farms, identify which issues presented the greatest challenges for U.S. tilapia farms, or assess the economic factors that contribute to the greatest costs on U.S. tilapia farms. Thus, there were no prior data from which to design a different type of study. A cross-sectional survey was done because the survey was to be done across all farms within the same time frame. Since farms do not maintain an accounting category of "regulatory costs," it was not possible to collect data over multiple years, as would be appropriate in a time-series analysis. A set of structured questions was developed from questions used in previous regulatory cost studies. Questions were modified as necessary for the tilapia sector based on review by knowledgeable Extension specialists, researchers, and pre-testing. The population to be studied was that of commercial tilapia producers in the U.S. The sampling design was that of a total census, in which every attempt was made to interview all known commercial tilapia producers in the U.S. The choice of a census rather than a random sample is consistent with the study design of previous regulatory cost studies in that previous studies revealed



substantial variability that necessitated conducting a census to avoid potential sample or coverage biases.

The survey was conducted nationally, with a concentration on the three major tilapia-producing states of California, Florida, and North Carolina. Additional well-established tilapia farms were contacted across the United States, but given the low number of such farms in many states, specific states cannot be disclosed for confidentiality reasons. Contact lists were developed with assistance from industry representatives, extension specialists, and others familiar with U.S. tilapia farmers. Attempts were made to contact all known commercial producers identified as existing tilapia businesses. Individuals viewed as trusted sources were asked to contact tilapia producers to encourage participation prior to initiating contact. Initial contact was made by email to each farmer on the list to introduce the project, explain the purpose of the study, summarize the results of previous regulatory cost surveys in other sectors, and request an appointment. Details of how the confidentiality of farm information was to be handled were provided at the time of the initial contact. The initial contact also verified whether the recipient was actively engaged in business, defined for the purposes of this study as raising and selling tilapia commercially. Those raising tilapia for home consumption and for educational/research purposes only (i.e., schools, universities, prisons) were excluded.

After three email contacts with no response, telephone calls were made to introduce the interviewer and request an appointment, with two follow-up telephone contacts in the event of a lack of response. All completed interviews were conducted by telephone, given the COVID-related travel restrictions in effect at the time of the survey. Interviews with Spanish-speaking tilapia producers were conducted in Spanish. All data were maintained as confidential business data and were coded so that there was no type of identifiable information; thus, no ethical approval was required.

*2.2. Survey Instrument*

The structured questions used in the survey instrument were developed from regulatory cost survey questions that had been used successfully in previous surveys [23,31–34]. As in previous regulatory cost studies, basic farm information was collected related to the production systems used, the overall scope and scale of the business, the primary markets targeted, and the total costs of production. Respondents were asked to identify the greatest overall problems on their farms (i.e., labor, markets, disease), and then the regulations that were most problematic. All local, state, federal, and international regulatory filings reported by respondents were recorded in lists by governance level. Cost information was collected on the various permits required, the costs associated with required testing (effluents, diseases), the investment required by regulatory authorities for new structures or equipment on farms, the manpower (labor and management) time and cost spent on regulations, and the consultant and legal fees paid.

The survey instrument was reviewed by extension specialists and researchers familiar with tilapia production in the U.S., pre-tested, and revised as necessary. A Spanish version of the survey instrument was developed for interviews with tilapia farmers who preferred to respond in Spanish.

*2.3. Data Analysis*

The data were analyzed by tallying the costs for each individual observation. The presentation of study results was based on formatting conventions developed in previous regulatory cost studies. Total production costs and total marketing costs were first summed for each observation. The costs related to regulations were then tabulated by disaggregating them from total production and marketing costs and further categorizing them into regulatory costs of permits and licenses, direct costs other than permits and licenses, labor and management costs associated with regulatory compliance, and additional investment and interest costs associated with regulatory compliance. The percentage of regulatory costs that were fixed versus variable was calculated. To identify which types of regulations

were most costly on tilapia farms, regulatory cost and revenue effects were divided into categories of effluent discharge, bird predation, water access, labor, taxes, fish health testing, processing, and transportation. Regulatory costs were calculated per state, per farm, and per kg of tilapia produced where there were sufficient observations to do so and maintain confidentiality. California, Florida, and North Carolina results were reported separately, while states with fewer than three responses were combined into an "other states" category. The national regulatory cost burden on U.S. tilapia farms was calculated by summing the values by state and adjusting for the coverage rate (average of percent coverage by weight and percent coverage by sales). All previous studies of the costs of regulation in other sectors of U.S. aquaculture showed substantially greater costs per kg of fish produced on smaller farms as compared to larger farms. In this analysis, the data were sorted by production volume into size categories of <100,000 kg/year, 100,000 to 200,000 kg/year, and >200,000 kg/year, and regulatory costs were compared across farm sizes.

Revenue lost because of regulations was found to be substantial in the previous salmonid, Pacific Coast shellfish, and Florida tropical studies [23,32,33]. Lost revenue is distinct from production and marketing costs. Lost revenue consists of sales revenue foregone from markets that were closed to aquaculture producers because of regulatory action, production lost because of specific regulations, attempted farm expansion thwarted by regulatory action or inaction, and lost business opportunities from the inability to acquire necessary permits. Additional questions elicited information on markets and production volumes that had been lost because of regulations, thwarted expansion attempts, and lost business opportunities that were a result of regulations. Lost revenue was categorized into lost markets, lost production, thwarted expansion, and lost business opportunities. The distinction between "lost markets" and "lost business opportunities" was whether respondents had records of previous sales to markets that had been lost to regulations ("lost markets") or markets that respondents believed they could access if it were not for regulations ("lost business opportunities") but for which there were no records of previous sales. Both categories of lost revenue were tabulated by regulatory category and by farm size.

## 3. Results

Survey respondents represented 74% of the volume (kg) and 76% of tilapia sales ($) in the U.S., for an average coverage rate of 75% (Table 1). The response rate (number of farm respondents as a percent of the number of tilapia farms listed in the 2018 Census) was low, at 18%. The high coverage rate, however, indicates that the survey dataset represents a very high percentage of all U.S. tilapia production. While the 2018 Census of Aquaculture lists 137 tilapia farms, the list frame developed for this study included only 44 farms that were identified as known commercial farms in business at the time of the survey (2021). Lists provided by contacts in various states included hobbyists, prisons, and school programs that were excluded from the list frame. Moreover, some state contacts reported several producers having gone out of business. Nevertheless, response and coverage rates were calculated based on the numbers and sales of tilapia reported in the 2018 Census of Aquaculture because there was no standardized method to judge the accuracy of the list frame developed for the survey. The response rate, in particular, is likely under-estimated as a result. By comparison, coverage rates for previous regulatory cost studies ranged from 37% to 100%, with response rates from 17% to 63% [23,31–34].

**Table 1.** List frame development, tilapia regulatory cost survey, 2021.

| State/Region | List Frame | No Response (Farms) | Completed Interviews (Farms) | Response Rate | Coverage Rate [a] |
|---|---|---|---|---|---|
| California | 8 | 1 | 7 | 87.5% | 75% |
| Florida | 17 | 11 | 6 | 35% | 61% |
| North Carolina | 8 | 3 | 5 | 62.5% | 73% |
| Other states [b] | 11 | 5 | 6 | 55% | 89% |
| National [c] | 137 | 20 | 24 | 18% | 75% |

[a] Averaged coverage rates by sales and by volume. [b] Includes five states with tilapia production, but too few tilapia farms in the state to disclose names of those states. [c] 2018 Census of Aquaculture lists 137 tilapia farms. The list frame developed for this study (44 farms) focused on known, commercial farms. Lists provided by various contacts included hobbyists, prisons, and school programs that were excluded from the contact list used for this study. Nevertheless, the numbers and sales of tilapia as reported in the 2018 Census of Aquaculture were used in calculating response and coverage rates despite the likely under-estimation.

*3.1. Characterization of U.S. Tilapia Production*

Respondents reported a wide variety of production systems used to raise tilapia, including ponds, raceways, and tanks (outdoor and indoor) (Table 2). Tank production included both flow-through and recirculating systems, both indoors and outdoors. Those with flow-through systems typically recycle water from fish production to other crops or through a reservoir with return flow to the tanks. Twenty-one percent of the farms used more than one production system to raise tilapia. Overall, half (50%) of all tilapia respondents raised tilapia in recirculating indoor tanks, but 13% also used flow-through tanks indoors. Of those using outdoor tanks, 21% were flow-through and 13% recycled water. Twelve percent of respondents used ponds, and 8% used raceways to produce tilapia. Farms in California and Florida raised tilapia in a variety of production systems, including ponds and outdoor and indoor tanks with both flow-through and recirculation of water, while all respondents in North Carolina used RAS (Table 2). The lack of a single, dominant production system contrasts sharply with the other major sectors of U.S. aquaculture. Catfish farms, for example, are dominated by open ponds that are transitioning to intensively aerated and split ponds [72], and the majority of trout are raised in raceways [32].

**Table 2.** Percent of farms with ponds, RAS, and outdoor tanks (flow-through and recirculating). Percentages in columns do not add to 100% because many tilapia farms used more than one type of production system.

| Production System | California | Florida | North Carolina | Other States | Total |
|---|---|---|---|---|---|
| **Ponds** | 28% | 34% | 0% | 0% | 12% |
| **Raceways** | 0% | 0% | 0% | 33% | 8% |
| **Outdoor tanks** | | | | | |
| Flow-through | 43% | 33% | 0% | 0% | 21% |
| Recirculating | 43% | 0% | 0% | 0% | 13% |
| **Indoor tanks** | | | | | |
| Flow-through | 14% | 33% | 0% | 0% | 13% |
| Recirculating | 14% | 33% | 100% | 67% | 50% |

The vast majority (99%) of tilapia raised in the U.S. were sold as live fish into retail supermarkets equipped with aquaria and tanks for holding live fish. A very small percentage (~1%) was sold for pond stocking. Most tilapia farms sold their fish to a wholesaler/distributor who purchased and loaded the fish at the farm and then transported the tilapia to markets for sale. Only a few tilapia farms had hauling trucks and delivered tilapia directly to retail markets. Tilapia is the only food fish sector in the U.S. for which nearly all its production is sold as a live food fish to end consumers. Other food fish species are sold

primarily to a processor, a wholesaler, or a distributor who processes and further packages the fish for subsequent sale to retail or restaurant markets.

### 3.2. Top Problems and Regulations

Respondents rated increasing costs as their top problem, primarily those related to feed, electricity, and water (Figure 3). Each bar represents the number of respondents who reported each type of problem as one of their top five. Problems associated with markets were the next greatest problem overall among the top five problems, followed by regulations and diseases. Brownouts, or partial electrical outages, and reliability of utilities (especially electricity) were the fifth-greatest problem, followed by labor, supply chain and distribution problems, and the ability to discharge water. While the ability to discharge water was categorized as a discharge regulation by respondents, the inability to discharge was driven in most cases by concerns related to the escape of non-native tilapia into the wild.

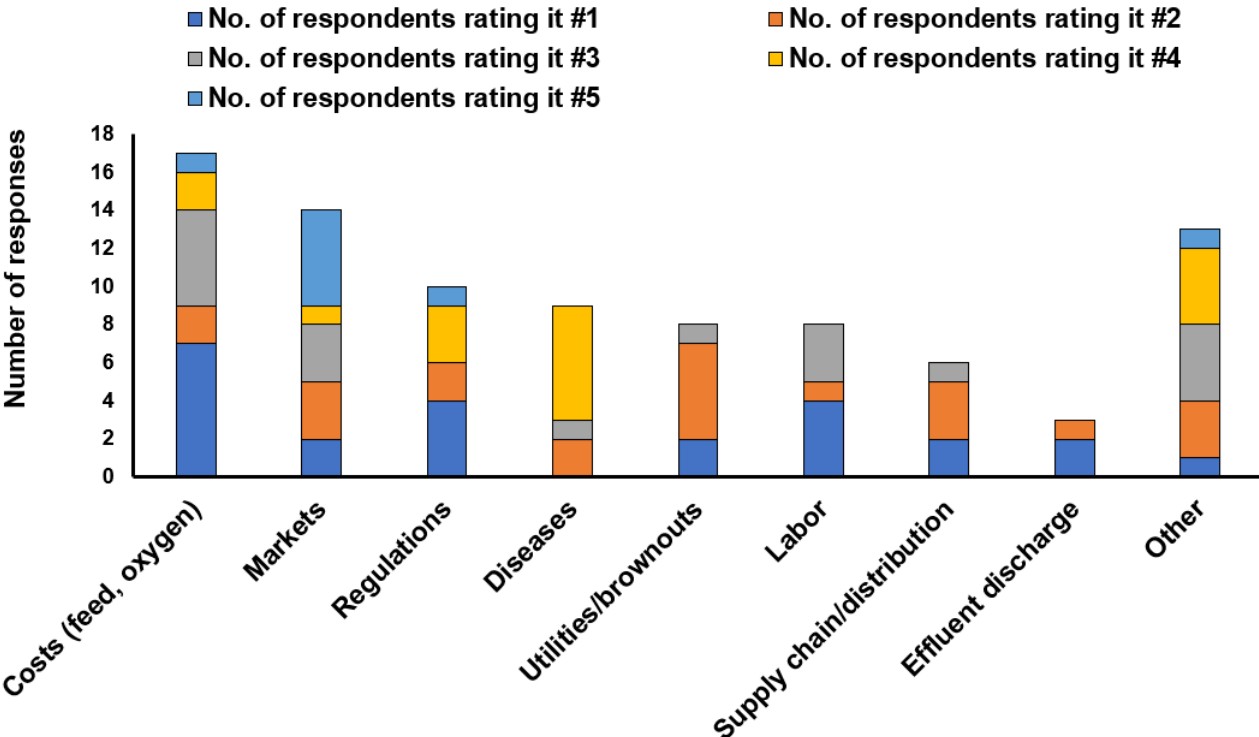

**Figure 3.** Producer responses to the following question: "We would like to put the importance of regulatory effects in the context of your overall business. What would you say are the top 5 biggest problems for your business? Please rank your top 5 problems with #1 being the biggest problem." (*n* = 24 respondents).

Of the various regulatory challenges identified by tilapia producers, effluent discharge was the most problematic, with the greatest number of respondents indicating that it was the #1 greatest problem (Figure 4). The international shipping and trucking category (including international permits to sell live fish, regulations related to electronic logs, and other trucking regulations) was second overall in terms of the top-five rankings, but problems with management of bird predation received more first-place rankings than international shipping. Policies related to water access and electricity were ranked third among regulatory problems, followed by state import health certifications, new/aquatic nuisance species, and drug approval/investigational new animal drug (INAD) processes.

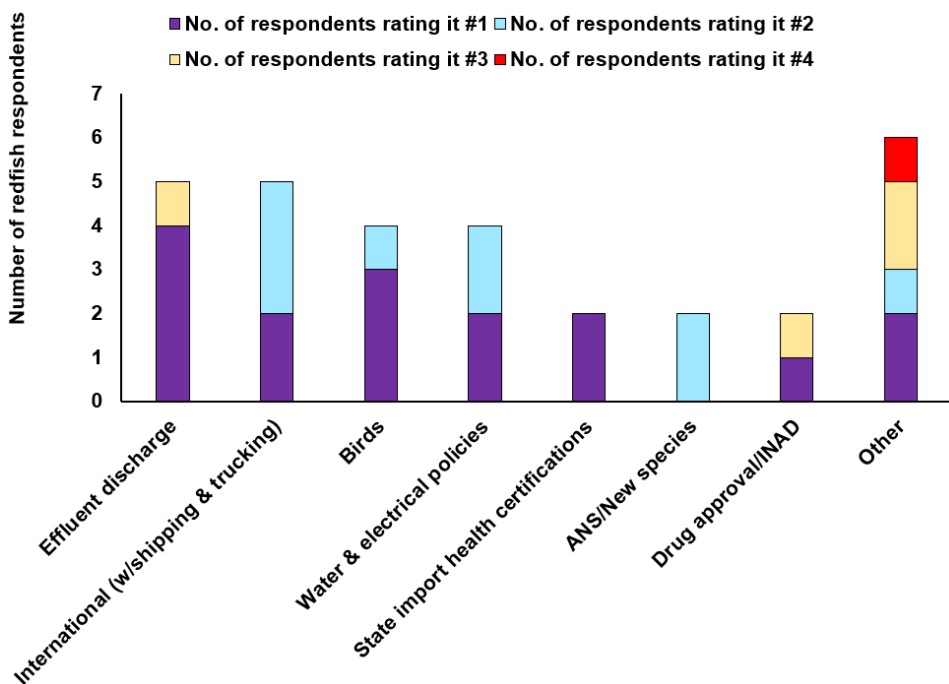

**Figure 4.** Producer responses to the following question: "Of all the regulations you deal with, which have created the greatest problems for your business, with #1 being the most problematic." (*n* = 24 respondents). "Other" includes: low-cost imports as barrier to entry, disease control (lack of approved antibiotics), FDA/processing regulations, OSHA, banking regulations, the cumulative effect of the total suite of regulations.

When asked if respondents received timely notification of annual renewals of permits and relevant changes in regulations, 42% of respondents indicated that they always received notification, and 46% indicated that they were always notified of changes in regulations (Table 3). Thirteen percent of respondents said that they never received reminders of the need for annual renewals, and 8% said that they never received notification of changes to regulations.

**Table 3.** Notification of annual renewals and changes in regulations. (*n* = 24).

| Rating (0 = Never; 5 = Always) | Annual Renewal (% of Respondents) | Notification of Changes in Regulations (% of Respondents) |
|---|---|---|
| 0 | 13% | 8% |
| 1 | 8% | 4% |
| 2 | 4% | 8% |
| 3 | 0% | 4% |
| 4 | 13% | 8% |
| 5 | 42% | 46% |
| No response | 21% | 21% |

Forty-two percent of respondents indicated that their business had experienced interruptions from regulatory delays related to required permits, and 38% reported that regulatory issues prevented them from expanding their businesses (Table 4). A third of respondents reported that they knew of tilapia farms that had gone out of business because of regulations; 29% said they had lost markets; and 21% said they had experienced unexpected changes to their business as a result of regulatory issues. In contrast with other sectors of U.S. aquaculture, 51% of salmonid farmers [32], 56% of Pacific Coast shellfish farmers [23], and 12% of catfish farmers [34] reported knowing of a farm that had gone out of business because of regulatory issues.

**Table 4.** Business interruptions, prevention of business expansion, farms that have gone out of business, and markets lost because of regulations (*n* = 24).

| Effect | Yes | No | No Response |
|---|---|---|---|
| Business interruptions | 42% | 54% | 4% |
| Prevention of expansion | 38% | 62% | 0% |
| Know of farms that have gone out of business | 33% | 50% | 17% |
| Lost markets | 29% | 71% | 0% |
| Unexpected changes | 21% | 67% | 12% |

*3.3. Number of Permits and Regulatory Filings*

Respondents reported 164 total regulatory filings (Table 5). A regulatory filing was defined as an activity required by regulatory agencies that involved a substantive study, survey, or other submission by the farm to obtain specific certificates or other approvals required as part of the permit application process. In addition to filing permit applications, other types of regulatory filings included engineering studies, wetland surveys conducted by hired consultants, and consultations required of tribal, coastal, or federal authorities. Routine submissions of water quality monitoring and testing, however, were not included as separate "filings." The numbers of filings varied from 15 (North Carolina) to 73 (other states), with the mean number per farm ranging from 3 to 12 (median = 3 to 9.5 per farm). Of the total regulatory filings, more than half (55%) were either state regulations or federally mandated state regulations. Twenty-five percent were locally promulgated and enforced regulations, while 19% were federal regulations.

**Table 5.** Number of regulatory filings [a] (includes all applications required).

| | California | Florida | North Carolina | Other States | Total |
|---|---|---|---|---|---|
| **Local** | | | | | |
| Total | 11 | 7 | 5 | 18 | 41 |
| Mean | 2 | 1 | 1 | 3 | 2 |
| Median | 2 | 1 | 1 | 1 | 1 |
| **State** | | | | | |
| Total | 18 | 21 | 10 | 31 | 80 |
| Mean | 3 | 3.5 | 2 | 5 | 3 |
| Median | 2 | 3 | 2 | 5 | 3 |
| **Federally mandated** | | | | | |
| Total | 3 | 2 | 0 | 6 | 11 |
| Mean | 0.4 | 0.3 | 0 | 1 | 0.5 |
| Median | 0 | 0 | 0 | 1 | 0 |
| **Federal** | | | | | |
| Total | 8 | 6 | 0 | 17 | 31 |
| Mean | 1 | 1 | 0 | 3 | 1 |
| Median | 0 | 1 | 0 | 2 | 0.5 |
| **International** | | | | | |
| Total | 0 | 0 | 0 | 1 | 1 |
| Mean | 0 | 0 | 0 | 0.2 | 0.04 |
| Median | 0 | 0 | 0 | 0 | 0 |
| **Total** | | | | | |
| Total | 40 | 36 | 15 | 73 | 164 |
| Mean | 6 | 6 | 3 | 12 | 7 |
| Median | 4 | 5.5 | 3 | 9.50 | 5 |

[a] A regulatory filing was defined as an activity required by regulatory agencies that required a substantive study, survey, or other submission by the farm to obtain specific certificates or other approvals required as part of a permit application process. Examples include engineering studies, wetland surveys conducted by hired consultants, or consultations required of tribal, coastal, or federal authorities. Routing submissions of water quality monitoring and testing, however, were not included as separate "filings."

The majority of regulatory filings were for various environmental management regulations or legal/labor regulations (Table 6). More than three-fourths (79%) of environmental management regulatory filings were in response to state or federally mandated state regulations. In comparison, the vast majority (73%) of legal/labor filings were at the local level, while only 6% of environmental management filings were local regulations. Most of the regulatory filings related to aquaculture permits (100%), water access (82%), and aquatic animal health (89%) were required at the state level, whereas interstate transportation filings and food safety regulatory filings were at the federal level.

**Table 6.** Total number of permits/filings by six regulatory categories and level of government responsible. Percentage values calculated by column.

| | Environmental Management | Legal/ Labor | Aquaculture Permit | Water Access | Interstate Transport | Aquatic Animal Health | Food Safety |
|---|---|---|---|---|---|---|---|
| **Local** | | | | | | | |
| No. | 3 | 35 | 0 | 3 | 0 | 0 | 0 |
| % | 6% | 73% | 0 | 18% | 0 | 0 | 0 |
| **State** | | | | | | | |
| No. | 27 | 7 | 24 | 14 | 0 | 8 | 0 |
| % | 56% | 15% | 100% | 82% | 0 | 89% | 0 |
| **Federally mandated** | | | | | | | |
| No. | 11 | 0 | 0 | 0 | 0 | 0 | 0 |
| % | 23% | 0 | 0 | 0 | 0 | 0 | 0 |
| **Federal** | | | | | | | |
| No. | 7 | 6 | 0 | 0 | 14 | 0 | 4 |
| % | 15% | 12% | 0 | 0 | 100% | 0 | 100% |
| **International** | | | | | | | |
| No. | 0 | 0 | 0 | 0 | 0 | 1 | 0 |
| % | 0 | 0 | 0 | 0 | 0 | 11% | 0 |
| **TOTAL** | | | | | | | |
| No. | 48 | 48 | 24 | 17 | 14 | 9 | 4 |
| % | 100% | 100% | 100% | 100% | 100% | 100% | 100% |

*3.4. Direct Regulatory Costs*

3.4.1. National and State Regulatory Costs

The national total regulatory costs for tilapia were $4.4 million (Table 7), as compared to $45.4 million for catfish [34], $16.1 million for salmonids [32], $15.6 million for Pacific Coast shellfish [23], and $5.2 million for Florida tropical fish [33]. The individual state with the greatest regulatory cost on tilapia farms was California, at $1.6 million, followed by "other states," Florida, and North Carolina. Per-farm, regulatory costs were $137,611 nationally, $226,788 in California, $80,462 in Florida, and only $28,854 in North Carolina. Per-kg of production (averaged across the total weight of tilapia produced), regulations added $0.66/kg, but this value ranged from a low of $0.39/kg in the "other states" category to $1.10/kg in California. As a percentage of total costs, regulatory costs on U.S. tilapia farms accounted for 15% of all costs of production nationally, with a range of 7% to 18%.

**Table 7.** Total regulatory cost by state, ($/state, $/farm, $/kg (averaged by total production and averaged by farm).

| State | $/State [a] | $/Farm (Mean) | $/kg (Averaged by Total Production) [b] | $/kg (Averaged by Farm) [c] | Regulatory Costs as % of Total Costs |
|---|---|---|---|---|---|
| California | $1,587,515 | $226,788 | $1.10 | $1.03 | 18% |
| Florida | $482,773 | $80,462 | $1.05 | $21.36 | 18% |
| North Carolina | $144,270 | $28,854 | $0.46 | $0.46 | 7% |
| Other states | $1,088,107 | $181,351 | $0.39 | $1.60 | 13% |
| Total respondents | $3,302,666 | $137,611 | $0.66 | $0.95 | 15% |
| National, coverage adjusted [d] | $4,403,554 | $137,611 | $0.66 | $0.95 | 15% |

[a] Calculated as sum of all regulatory costs across observations from that state. [b] Calculated by dividing the $/state by the total weight of production reported for that state. [c] Calculated by dividing the regulatory cost per kilogram of production on each farm in that state and then averaging the per-kg regulatory cost across all observations in that state. [d] Coverage was 74% of total tilapia sales and 76% of total weight of tilapia sold; 75% was used to adjust for coverage to estimate national values.

### 3.4.2. Regulatory Costs by Type of Cost

Regulatory compliance resulted in a variety of different types of costs on tilapia farms. The greatest of these was taxes, at 28% of total regulatory costs, followed by capital costs (20%), manpower (18%), water costs (5%), legal, accounting, and professional services (4%), with permits and licenses accounting for only 1% of total regulatory compliance costs (Figure 5). Direct costs other than labor constituted 24% of the total costs of regulation.

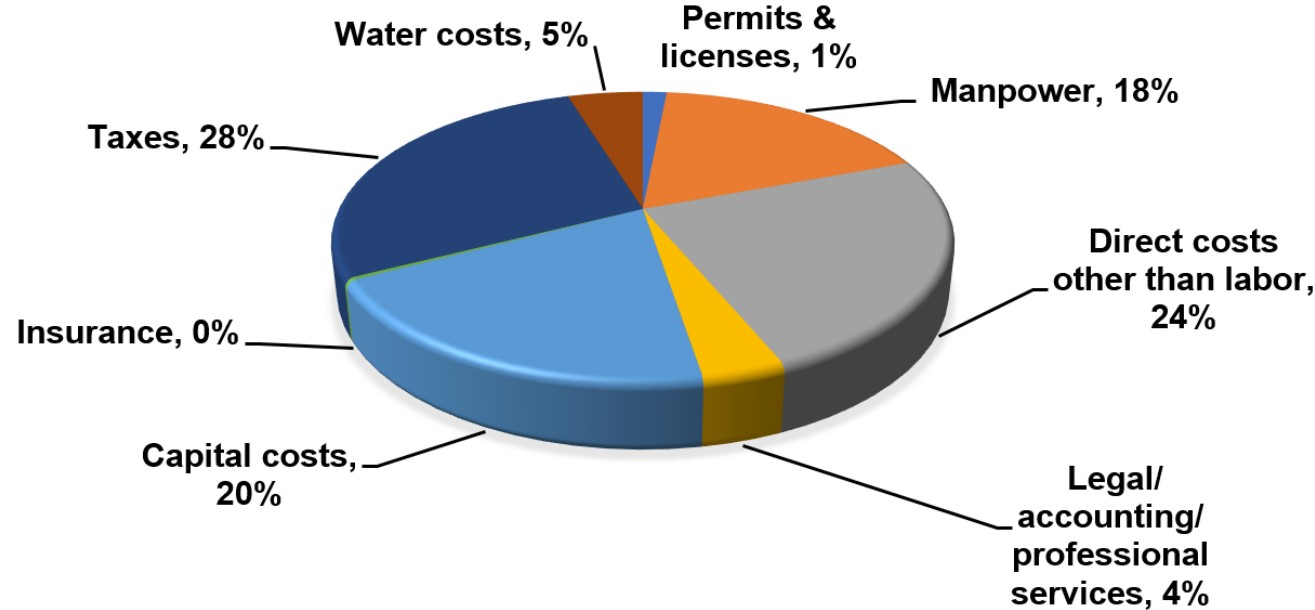

**Figure 5.** Regulatory costs by type of cost (% of regulatory costs).

The relative importance of different cost categories varied by state. For example, in California, manpower costs were the greatest cost created by regulations (Table 8). The greatest type of regulatory cost for Florida and the other states, however, was the sum of direct costs other than permits. The only regulatory costs reported by North Carolina respondents were taxes paid, a small amount of interest on capital, and manpower.

**Table 8.** Total regulatory costs (by type of cost) by farm and by state (Values in mean $/farm).

| | California | Florida | North Carolina | Other States | National |
|---|---|---|---|---|---|
| Permits & licenses | $5794 | $742 | $0 | $727 | $2057 |
| Direct costs other than permits | $5362 | $36,804 | $0 | $91,627 | $33,672 |
| Manpower | $74,980 | $6470 | $38 | $3698 | $24,419 |
| Water costs | $22,300 | $0 | $0 | $0 | $6504 |
| Legal/accounting/ professional services | $13,196 | $0 | $0 | $4947 | $5086 |
| Capital costs | $52,677 | $27,996 | $2620 | $17,363 | $27,250 |
| Insurance | $1301 | $0 | $0 | $0 | $379 |
| Taxes | $51,178 | $8450 | $26,196 | $62,989 | $38,244 |

### 3.4.3. Regulatory Costs by Category of Regulation

The categories of regulations that contributed the most to regulatory costs were those related to discharge of effluents (20% of total regulatory costs), control of predatory birds (14%), non-native or aquatic nuisance species (ANS) (10%), water access (7%), and fish health (2%) (Figure 6). The greatest percentage of regulatory costs was in the "all other regulatory costs" category, with 47% of all regulatory costs. Examples of the costs grouped into the "all other regulatory costs" category (because of the limited number of observations) included farms with high costs for guest worker visas, increased feed costs related to drugs under the investigative new animal drug (INAD) programs, consulting and other professional services, taxes, and increased interest on operating and investment capital.

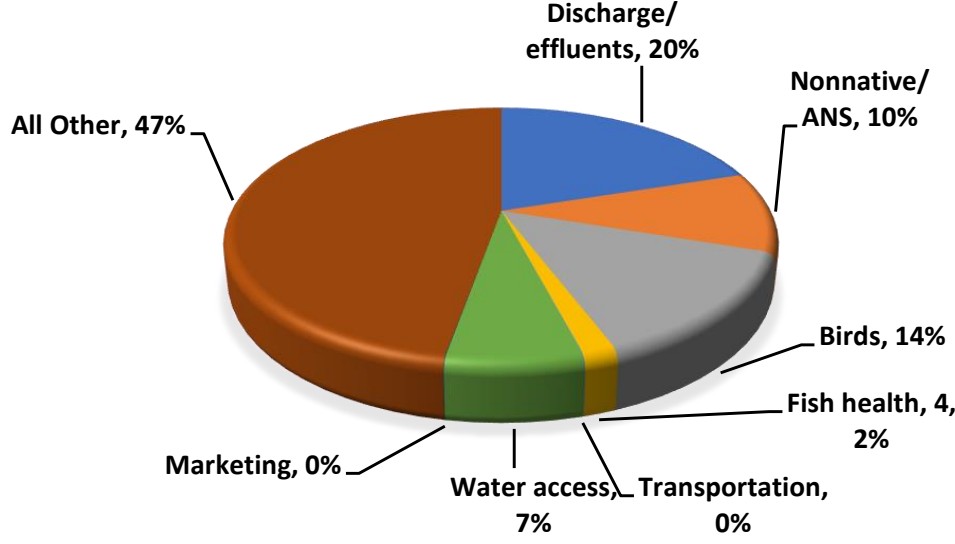

**Figure 6.** Regulatory cost by type of regulation (% of regulatory cost).

State-to-state variation was evident in the relative magnitude of costs from different regulatory categories (Table 9). For example, in California, the costliest regulatory category was that related to non-native species, followed by management of predatory birds, and then water access. In Florida, predatory birds resulted in the greatest costs, but effluent discharge was the greatest regulatory cost in the "other states" category. Per-kg of production, effluent discharge regulations cost from $0 to $0.206/kg; non-native species regulations cost from $0 to $0.217/kg and management of bird predators cost from $0 to $0.462/kg (Table 10).

**Table 9.** Total regulatory cost by regulatory category by state ($/farm).

|  | California | Florida | North Carolina | Other States | National |
|---|---|---|---|---|---|
| Effluent discharge | $7943 | $975 | $0 | $94,985 | $26,307 |
| Nonnative species | $44,814 | $838 | $0 | $341 | $13,365 |
| Birds | $32,318 | $35,451 | $0 | $966 | $18,530 |
| Fish health | $0 | $0 | $0 | $8888 | $2222 |
| Transportation | $183 | $0 | $0 | $25 | $60 |
| Water access | $30,026 | $459 | $0 | $420 | $8977 |
| Marketing | $0 | $0 | $0 | $192 | $48 |
| All other | $111,503 | $42,740 | $28,854 | $75,534 | $68,102 |
| Total | $226,788 | $80,462 | $28,854 | $181,351 | $137,611 |

**Table 10.** Total regulatory cost by regulatory category by state ($/kg, mean of total production).

|  | California | Florida | North Carolina | Other States | National |
|---|---|---|---|---|---|
| Effluent discharge | $0.038 | $0.013 | $0 | $0.206 | $0.126 |
| Nonnative species | $0.217 | $0.011 | $0 | $0.001 | $0.064 |
| Birds | $0.156 | $0.462 | $0 | $0.002 | $0.089 |
| Fish health | $0 | $0 | $0 | $0.019 | $0.011 |
| Transportation | $0.001 | $0 | $0 | $0.0001 | $0.0003 |
| Water access | $0.145 | $0.006 | $0 | $0.001 | $0.043 |
| Marketing | $0 | $0 | $0 | $0.0004 | $0.0002 |
| All other | $0.539 | $0.558 | $0.457 | $0.164 | $0.327 |
| Total | $1.097 | $1.050 | $0.457 | $0.393 | $0.661 |

### 3.4.4. Regulatory Costs by Farm Size

While regulatory costs per farm were significantly ($p < 0.05$) greater on larger farms, when calculated on a per-kg basis (averaged across farms), the regulatory cost per kg was significantly lower on larger farms (Table 11). Moreover, chi-square tests revealed that the percentage distribution of regulatory costs across regulatory categories differed significantly by farm size. Management of predatory birds accounted for the greatest percentage of total regulatory costs on farms in the smallest size group. On medium-sized farms, effluent discharge regulations comprise the greatest regulatory costs. In the largest farm size category, more than half (58%) of regulatory costs were in the "all other regulatory costs" category.

**Table 11.** Farm size effects of regulatory costs.

| Metric | <100,000 kg/year | 100,000 to 200,000 kg/year | >200,000 kg/year |
|---|---|---|---|
| **Regulatory costs ($)** |  |  |  |
| $/farm | $49,963 | $221,559 | $264,757 |
| $/kg, averaged by farm | $10.35 | $1.67 | $0.54 |
| $/kg, weighted by total kg production | $1.11 | $1.69 | $0.37 |
| **Regulatory costs (% of total regulatory costs)** |  |  |  |
| Discharge/effluent | 2% | 44% | 2% |
| Nonnative species | 0% | 1.5% | 23% |
| Birds | 21% | 8% | 16% |
| Fish health | 8% | 0% | 0% |
| Transportation | 0% | 0% | 0% |
| Water access | 1% | 14% | 1% |
| All other | 68% | 32% | 58% |

### 3.5. Lost Revenue from Regulations

3.5.1. Lost Revenue Nationally and by State

Lost revenue had a much greater economic effect on tilapia farms than did the increased costs resulting from regulations. Nationally, the total revenue lost from regulations was eight times greater than that of direct regulatory costs, at $32 million annually (Table 12). The largest portion of the lost revenue ($23.4 million) was experienced by tilapia farms in the "other states" category. Lost revenue resulted from three different effects that included the value of lost production, lost markets and business opportunities, and thwarted attempts to expand the business. Of these different effects, the largest category of lost revenue was that of thwarted attempts to expand the business, at $16.7 million in annual lost revenue. The greatest amount of lost production was reported from Florida, where tilapia producers were no longer able to discharge water because of regulations related to the potential for escape of non-native tilapia. Many Florida tilapia farms did not have sufficient land to construct reservoirs to hold discharge water and had been forced to reduce stocking densities, which resulted in reduced tilapia yields. Lost business opportunities were reported to have resulted from labor shortages and difficulties with processing visas for immigrant labor that prevented farms from taking advantage of various business opportunities. Much of the thwarted expansion was because of the difficulty in navigating approvals and required permits to develop processing capabilities. The lack of processing capacity prevents tilapia farms from expanding into the much larger fillet market in the U.S.

**Table 12.** Lost revenue from lost production, thwarted expansion, and lost opportunities.

| | California | Florida | North Carolina | Other States | National |
|---|---|---|---|---|---|
| **Lost production** | | | | | |
| $/state [a] | $57,000 | $2,553,229 | $0 | $504,800 | $3,115,029 |
| $/farm (mean) | $7200 | $425,538 | $0 | $63,100 | $129,793 |
| $/kg, averaged per farm [b] | $0.052 | $10.97 | $0 | $27.25 | $9.57 |
| $/kg, per total kg produced [c] | $0.039 | $5.55 | $0 | $0.182 | $0.62 |
| **Thwarted expansion** | | | | | |
| $/state [a] | $0 | $250,000 | $0 | $16,450,000 | $16,700,000 |
| $/farm (mean) | $0 | $41,667 | $0 | $3,290,000 | $726,087 |
| $/kg, averaged per farm [b] | $0 | $0.184 | $0 | $28.53 | $7.18 |
| $/kg, per total kg produced [c] | $0 | $0.544 | $0 | $5.94 | $3.34 |
| **Lost opportunities** | | | | | |
| $/state [a] | $3,549,000 | $0 | $0 | $601,502 | $4,150,502 |
| $/farm (mean) | $507,000 | $0 | $0 | $85,929 | $172,938 |
| $/kg, averaged per farm [b] | $2.06 | $0 | $0 | $0.255 | $0.84 |
| $/kg, per total kg produced [c] | $2.45 | $0 | $0 | $0.416 | $0.83 |
| **Total respondents** | | | | | |
| $/state [a] | $3,606,000 | $2,803,229 | $0 | $17,556,302 | $23,965,531 |
| $/farm (mean) | $515,143 | $467,205 | $0 | $2,926,050 | $998,564 |
| $/kg, averaged per farm [b] | $2.11 | $11.16 | $0.00 | $28.79 | $10.60 |
| $/kg, per total kg produced [c] | $2.49 | $6.09 | $0.00 | $6.34 | $4.80 |
| **National (adjusted for coverage) [d]** | | | | | |
| $/state [a] | $4,808,000 | $3,737,639 | $0 | $23,408,403 | $31,954,042 |
| $/farm (mean) | $515,143 | $467,205 | $0 | $2,926,050 | $998,564 |
| $/kg, averaged per farm [b] | $2.11 | $11.16 | $0.00 | $28.79 | $10.60 |
| $/kg, per total kg produced [c] | $2.49 | $6.09 | $0.00 | $6.34 | $4.80 |

[a] Calculated as sum of all regulatory costs across observations from that state in $ (USD). [b] Calculated by dividing the regulatory cost per kilogram of production on each farm in that state and then averaging the per-kg regulatory cost across all observations in that state. [c] Calculated by dividing the $/state by the total weight of production reported for that state. [d] Coverage was 74% of total tilapia sales and 76% of total weight of tilapia sold; used 75% to estimate national values.

Overall, total lost revenue was 82% of total sales revenue on tilapia farms (Table 13). Lost revenue exceeded total sales of tilapia in "other states," including 95% of total sales in Florida and 39% of total sales in California. By comparison with previous regulatory cost studies of other U.S. sectors, total annual lost revenue for salmonids was $52.5 million [32], $23.2 million for Florida tropical fish [33], $35 million for catfish [34], and $169.9 million for Pacific Coast shellfish [23].

**Table 13.** Lost revenue as percentage of total sales.

|  | California | Florida | North Carolina | Other States | National |
|---|---|---|---|---|---|
| **Total lost revenue as percent of total sales** | 39% | 95% | 0% | 116% | 82% |
| **Categories of lost revenue** | | | | | |
| Market opportunities foregone as percent of total sales | 32% | 0% | 0% | 0% | 10% |
| Lost production as percent of total sales | 1% | 86% | 0% | 3% | 11% |
| Thwarted expansion as percent of total sales | 0% | 8% | 0% | 108% | 57% |
| Lost opportunities as percent of total sales | 6% | 0% | 0% | 4% | 4% |

### 3.5.2. Lost Revenue by Category of Regulation

By category of regulation, the greatest amount of lost revenue per farm was that of regulations related to non-native species, followed by those related to effluent discharge, labor, managing predatory birds, fish health, and water access regulations (Table 14). The most important sources of lost revenue, however, varied by state. In California, labor regulations accounted for the greatest mean value of lost revenue per farm, followed closely by non-native species regulations, whereas regulations on the discharge of effluent resulted in the greatest amount of lost revenue per farm in Florida. For the "other states" category, regulatory challenges associated with processing resulted in the greatest lost revenue, followed by labor regulations.

**Table 14.** Lost revenue by type of regulations, in $/farm.

|  | California | Florida | North Carolina | Other States | National |
|---|---|---|---|---|---|
| Effluent discharge | $0 | $648,910 [a] | $0 | $0 | $114,513 |
| Non-native species | $423,429 | $0 | $0 | $2333 | $124,083 |
| Birds | $8143 | $121,300 | $0 | $1600 | $33,415 |
| Labor | $585,000 | $0 | $0 | $293,750 | $83,750 |
| Fish health | $0 | $0 | $0 | $83,333 | $20,833 |
| Water access | $0 | $41,667 | $0 | $0 | $5682 |
| All other | $0 | $0 | $0 | $3,290,000 | $74,772 |
| Total | $515,143 | $467,205 | $0 | $2,926,050 | $998,564 |

[a] Tilapia are non-native, and discharge is not allowed to prevent escape.

### 3.5.3. Lost Revenue by Farm Size

The largest farms accounted for the majority of lost revenue from thwarted expansion attempts (Table 15). Examples of thwarted expansion attempts included the inability to obtain permits for additional wells; develop processing capacity to compete in the tilapia fillet market, given regulatory and other cost differentials with exporting nations; and obtain permits for non-native species production. Business opportunities have been lost because of delays in states approving import permits caused by differing interpretations of state statutes and rules.

**Table 15.** Farm size effects of lost revenue.

| Metric | <100,000 kg | 100,000 to 200,000 kg | >200,000 kg |
|---|---|---|---|
| **Value of thwarted expansion** | | | |
| $/farm | $0 | 0 | $3,340,000 |
| $/kg, averaged by farm | $0 | 0 | $1.76 |
| $/kg, weighted by total kg production | $0 | 0 | $4.61 |
| **Value of lost production** | | | |
| $/farm | $89,551 | $118,133 | $248,413 |
| $/kg, averaged by farm | $16.82 | $0.93 | $1.09 |
| $/kg, weighted by total kg production | $1.99 | $0.90 | $0.34 |
| **Value of lost opportunities** | | | |
| $/farm | $1077 | $260,000 | $234,500 |
| $/kg, averaged by farm | $0.02 | $1.46 | $0.59 |
| $/kg, weighted by total kg production | $0.02 | $1.98 | $0.32 |
| **Value of total lost revenue** | | | |
| $/farm | $90,628 | $378,133 | $3,822,913 |
| $/kg, averaged by farm | $16.84 | $2.39 | $3.45 |
| $/kg, weighted by total kg production | $2.01 | $2.88 | $5.28 |

Lost production and lost opportunities were reported by all farm size categories. The types of regulations that caused the greatest loss of revenue varied by farm size (Table 16). On the smallest farms, management of predatory birds accounted for the greatest percentage of lost revenue. On medium-sized farms, the greatest lost revenue was from the regulation of tilapia as a non-native species. In the largest farm size category, nearly all lost revenue (86%) was in the "all other" regulatory category.

**Table 16.** Farm size effects of lost revenue by regulatory cost category (values in % of total lost revenue).

| Regulatory Category | <100,000 kg | 100,000 to 200,000 kg | >200,000 kg |
|---|---|---|---|
| Discharge/effluents | 10% | 29% | 6% |
| Non-native species | 1% | 69% | 0% |
| Birds | 46% | 2% | 0.4% |
| Fish health | 42% | 0% | 0% |
| Water access | 0% | 0% | 1% |
| Labor | 0% | 0% | 6% |
| All other | 0% | 0% | 86% |

## 4. Discussion

Continued growth of the global population has led to dual, but inter-related challenges of how to provide a safe and healthy food supply while simultaneously improving social and environmental quality. An effective regulatory framework is necessary to reduce externalities associated with the use of common property resources [2] while supporting economic sustainability. A growing body of research has emerged that provides evidence that in developed countries such as the U.S., the governance framework may be constraining the growth of newer, promising economic sectors such as aquaculture [20,27]. Research evidence has suggested that the disparity in environmental regulations has contributed to the shift of aquaculture production from developed countries such as the U.S. and the EU to developing countries where there is inadequate control of negative externalities from aquaculture production. Costs of aquaculture production are lower in developing countries, in large part because of the discrepancy in social and environmental quality standards [73].

The costs of regulations have been measured in the major sectors of U.S. aquaculture, except for tilapia. This study takes a step to fill the knowledge gap about the regulatory

costs of the U.S. tilapia sector. Tilapia provides a unique contrast in that its production has contracted in the U.S. while tilapia production globally has continued to grow, particularly in developing nations, many of which target the U.S. for tilapia exports. However, the economic cost differential that results from the lack of equivalence in regulatory stringency and costs may be incentivizing export-oriented production in developing countries to the detriment of both food security and environmental quality.

The economics of the U.S. tilapia industry has not been well studied, despite commercial farms that date back to the mid-1970s and a substantial amount of research in the U.S. on tilapia genetics, nutrition, and production technologies. This study has documented a strikingly different industry structure than that of other major U.S. aquaculture sectors. Rather than a cluster of farms in a single geographic region, as in the U.S. catfish industry [74], tilapia farms are spread across the U.S., with several, smaller clusters in several states. Some of the largest tilapia farms are not associated with clusters but have developed in relative isolation. Moreover, U.S. commercial tilapia farms exhibit substantial variation in production systems, from open ponds to RAS, and a measurable proportion makes use of outdoor tanks.

Tilapia production is especially important to the states of California, Florida, and North Carolina, accounting for 28% of all food fish sales in California and 11% of all aquaculture, 39% of all food fish sales and 2% of total aquaculture sales in Florida, and 10% of all food fish sales and 9% of all aquaculture sales in North Carolina. The large tilapia farms outside these three main cluster areas account for a high percentage of the food fish and of all aquaculture in their respective states.

Analysis of the economic effects of regulations on U.S. tilapia farms in this study shows that direct costs of regulation nationally contributed 15% of the total costs of production, greater than the 8.5% and 12% found on catfish and trout farms, respectively [32,34]. As found in previous studies, regulatory costs varied substantially by state, from 7% in North Carolina to 18% in California and Florida. While all tilapia farms in North Carolina used RAS, RAS farms in California and Florida demonstrated substantial regulatory costs. Thus, it is state-level approaches to regulation that drive the cost differences, not the production system. Regulatory costs were similarly found to be highest in California among Pacific Coast shellfish-producing states [23].

The costs of the time spent by labor and management for record-keeping, reporting, and meetings were one of the greatest costs of regulatory compliance on tilapia farms, measured at 18%, compared to 23% in salmonids, 11% on baitfish/sportfish farms, and 38% on catfish farms [23,32,34]. The cumulative burden of time and paperwork spent on regulatory issues is underappreciated. Farmers typically do not receive salaries, and time spent on regulatory paperwork is time not spent on farming. Greater regulatory compliance costs were found to be significant in reducing technical efficiency on U.S. baitfish/sportfish farms [31].

Consistent with other U.S. aquaculture sectors studied, the cost of permits and licenses was a very small portion of the overall regulatory compliance burden, at 1% for tilapia farms. For tilapia, water costs accounted for a measurable (5%) amount of the total direct costs of regulation, unlike findings from the previous regulatory cost studies. Regulatory costs of water reflect primarily water shortages and management by water districts in California to a much greater degree than in other states. Nevertheless, respondents outside of California indicated growing concerns over future availability, potential restrictions, and the increased cost of water for tilapia production.

Of the various specific regulatory categories, environmental management regulations accounted for the greatest costs on tilapia farms as was found in most other studies. Regulatory costs associated specifically with discharge of effluents from tilapia farms accounted for 20% of total regulatory costs, while those associated with management of predatory birds accounted for an additional 14% of total regulatory costs. In Florida, the discharge prohibition was related primarily to the prevention of tilapia escapes, given its non-native species status. In several cases, existing tilapia farms did not have additional

land on which to construct reservoirs for storage or treatment of effluents. In other areas, the surrounding land area was not available for sale, was inappropriate for reservoir construction, or was too expensive and not affordable for tilapia farms. Constraints to the availability of additional land on which to construct treatment systems for effluents were also identified that make on-farm effluent treatment an infeasible option for managing trout effluents [75]. The wide geographic spread of tilapia farms in the U.S. resulted in a large number of other regulatory constraints that were combined into an "all other category" that accounted for 47% of total regulatory costs. These other regulatory costs were associated with visas for immigrant labor, long-term INADs rather than formal drug approval, which resulted in high feed costs, local construction costs for a variety of structures, and market-related regulatory costs.

Previous studies on U.S. baitfish/sportfish and trout farmers, who primarily sell live fish, found that fish health testing costs were a major regulatory cost, but fish health costs were found to be low on tilapia farms (2% of total regulatory costs) [35,36]. The primary difference is that tilapia is sold as food fish, not for stocking into ponds or other water bodies. In addition, most tilapia farms sold fish to wholesalers and distributors, who picked up the fish at the pond bank; only a few tilapia farmers transported their own tilapia to markets. Wholesalers and distributors are reported to manage the permits necessary for transporting live fish.

The greatest economic effect of regulations on tilapia farms overall was that of lost revenue, which captured the effects of lost production, thwarted attempts to expand production, and lost opportunities. The value of lost revenue from regulations was equivalent to 82% of the total 2018 sales of U.S. tilapia farms. Similar constraining effects of regulations have been identified in all other regulatory cost studies. While the specific nature of the lost revenues varied across respondents and states, 70% of the lost revenue was from thwarted expansion attempts. Thus, tilapia farmers have been seeking to expand production to meet existing demand in live fish markets but have been unable to do so, largely because of the regulatory framework in the U.S. Restrictions on drilling new wells and the denial of permits to raise a non-native species thwarted attempts to expand production facilities, and increasing production costs. Difficulties in identifying permitting pathways for developing processing capacity increased the difficulty of competing in the tilapia fillet market. Production losses were reported from predatory birds, reduced stocking densities necessitated by restrictions on discharge that led to water quality problems, and the inability to obtain necessary tests for specific diseases required for the shipment of fish in a timely manner.

Regulatory effects varied with farm size, with the largest farms exhibiting the lowest per-kg regulatory costs but the greatest regulatory costs overall. The smallest farms had the greatest regulatory cost per kg of tilapia produced. Some large tilapia farms reported few regulatory issues for many years, but those regulatory costs were expected to increase mostly due to increasing discharge fees and monitoring requirements triggered by increased population growth and increased discharges by local communities to local waterways.

Regulatory issues vary over time in different locales with changes in state and local statutes and rules. This study captured only a snapshot of the regulatory costs identified in the single year of the study. There likely is a cumulative effect of regulations over time that was not captured in this study, and the values reported in this study may under-represent the total regulatory burden.

As with the previous studies on the economic effects of regulations on U.S. aquaculture, we measured only the costs and did not attempt to measure the benefits of regulations. There is no question that the regulatory framework in the U.S. has resulted in improved environmental quality, a safer workplace, and a safer food supply. Moreover, respondents to the survey were not suggesting or advocating for less stringent regulations but were reporting redundant, overlapping, and inefficient monitoring and reporting systems that resulted in higher costs than necessary. The finding that compliance with the total set of regulations on U.S. aquaculture has become one of the major costs of production points to

a strong need to search for "smarter" regulations [76], as suggested by survey respondents. More cost-effective regulations are needed, as suggested by various authors [4,35,36,77].

The U.S. tilapia industry sells almost exclusively into live food fish markets, which also differentiates this sector from the other food fish sectors studied to date (i.e., catfish and trout). While the live fish market is generally considered to be a limited market, there is some evidence of growth in modern supermarket chains that have invested in aquaria to hold live fish and that shoppers who purchase live seafood products are willing to pay a premium price [78]. Other justifications for projections of growth in live fish markets included: (1) the increasing size of the Asian population; (2) greater per-capita consumption of seafood by Asians [79]; and (3) greater buying power of Asians as compared with other U.S. demographic segments [80]. Other studies have characterized those who purchased live fish in the North Central [81,82] and Northeast live fish markets [83]. For tilapia, survey respondents did report that other species of fish, including trout, catfish, and largemouth bass, are being sold in greater volumes in live fish markets. Thus, substitution by customers of live fish markets for tilapia for other species being sold live may have contributed to some extent to the contraction of U.S. tilapia sales. Live fish markets, however, have continued to be quite small relative to the large retail and food service sales that consist primarily of fish fillets.

Substantial growth of the U.S. tilapia industry likely will depend upon its competitiveness in the large U.S. fish fillet market that is currently dominated by imported seafood sold typically at lower prices than domestically farmed product. With respect to tilapia, many industry observers believe that the lower dress-out percentage of tilapia fillets as compared with catfish and other species is an insurmountable impediment to competitiveness in the U.S. market. Nevertheless, several survey respondents reported having developed unique genetic strains of tilapia with faster growth that yield a greater fillet dress-out percentage. Moreover, new fillet machines designed to process tilapia are available on the market and reportedly yield higher dress-out percentages of tilapia fillets.

Can U.S. tilapia producers be competitive in the large U.S. fillet market? There are several factors that suggest that there may be a pathway for U.S.-farmed tilapia to move into fillet markets. U.S. consumers are now familiar with tilapia, but the low quality of imported frozen tilapia appears to have reduced consumer interest [60], which may offer opportunities for high-quality, U.S.-farmed tilapia. Regulatory reform offers opportunities to reduce the high regulatory burden on tilapia farms, which could result in a more competitive price for consumers. However, improving the competitiveness of the U.S. tilapia sector likely will require greater attention from U.S. research and extension scientists. A number of survey respondents reported frustrations related to the lack of research and extension support for their sector. Specific needs are for research and development of cost-efficient processing technologies, testing of new tilapia fillet equipment, and marketing research to provide guidance on effective market penetration and growth strategies for U.S. tilapia producers.

Limitations of this study include its focus only on tilapia production in the U.S. While the respondents to the survey represented 75% of all U.S. tilapia production, the respondent pool was limited to 24 respondents that were spread across a number of states. A useful extension of this work would be to measure the regulatory costs of tilapia in other countries, including sufficient data to disaggregate and compare across various levels of technological development. Such a cross-country analysis could provide a basis for a broader discussion of the global dynamics of tilapia production and the extent to which the disparity in regulatory costs and environmental management has driven the shift in aquaculture production to other parts of the world.

## 5. Potential Pathways for Growth of U.S. Tilapia Production

This study has suggested some potential pathways for the growth of U.S. tilapia production. The major findings of this study showed that the decline in U.S. tilapia

production may have resulted from two different factors: one is the regulatory structure and associated cost increases, and the second is access to the large U.S. market for food fish fillets.

This study highlighted the need to improve the efficiency of the regulatory framework in the U.S. for tilapia, particularly given that a pragmatic regulatory framework has been slow to develop in the U.S. [50]. Regulations in the U.S. have more often been of the command-and-control type, which has been shown to be less effective and efficient than incentive-based regulations [84].

In addition, reduced paperwork will reduce costs in the form of the value of personnel time spent on monitoring, record-keeping, and reporting. Reducing the time burden of reporting would free up time on aquaculture farms for innovation and other efficiency enhancements [77]. Regulatory agencies would further benefit from cost savings from the reduction of the time burden related to monitoring and reporting.

The advances in information technology in the past decades likely offer potential solutions for alleviating the costs of monitoring, record-keeping, and reporting. Dashboards could be developed for producers to upload all required monitoring data for all regulatory agencies. Centralizing such data at the level of a state agriculture department for all regulatory reporting would simplify and streamline the reporting requirements. For effluent monitoring, actively rewarding producers with a history of no non-conformities with reduced testing would reduce the time and expense required. For those farms that need on-farm treatment, programs to offer partial subsidies or matching funds to construct on-farm or other treatment options would be of value. For those farms for which fish health testing costs to ship to other states or countries, risk-based approaches have been shown to have potential to substantially reduce the on-farm regulatory compliance burdens [35,36].

In developed countries, farmers, scientists, and citizens support a regulatory framework that results in the internalization of environmental and social externalities through taxes paid to cover the costs of the regulatory agencies [19]. Farmers, economists, and many in the general public recognize the necessity and benefits of laws and regulations that internalize various externalities to achieve societal goals related to the environment, public health, and an orderly society. However, periodic training is needed for farm-level inspectors, permit writers, and other regulatory personnel to keep up with the rapidly developing new technologies being adopted by aquaculture producers.

In addition, there is a need for sunset clauses in all regulatory actions. The rapid evolution of aquaculture technologies has resulted in improved technologies, but inflexible regulatory processes impede their adoption [4]. To avoid the unintended negative consequences documented in this and other studies, the options are to either develop more adaptive and flexible approaches or to include sunset clauses that result in periodic re-evaluation of the relevance of regulations promulgated.

The interviews conducted as part of this study documented a disturbing trend among U.S. tilapia producers regarding the lack of access to extension support services across the various states where tilapia farms are located. Numerous studies have documented the substantial positive impacts of the U.S. Extension service on U.S. agriculture generally [85] (Wang 2014). In U.S. aquaculture [86], the contributions of research and extension support from land-grant and federal laboratories played a substantial role in overcoming technological bottlenecks to economic sustainability in the U.S. catfish industry. While the catfish, trout, shellfish, and other sectors in the U.S. have continued to benefit from research and development from universities and extension personnel, the tilapia sector has not. Therefore, one important pathway to growth for U.S. tilapia producers is for land-grant universities and the United States Department of Agriculture to re-commit to investing in tilapia aquaculture production methodologies and providing the R&D support needed to adapt new technologies on farms. There is an especially strong need for such R&D support to test and adapt new processing equipment for U.S.-raised tilapia. If feasible, as some producers believe it is, efficient processing equipment could potentially provide the basis for rapid expansion of U.S.-raised tilapia fillets into the larger seafood fillet market.

## 6. Conclusions

This study focused on measuring the on-farm economic effects of the regulatory environment on U.S. tilapia producers. Study results show that the magnitude of the economic effects is such that there is a need for regulatory reform to enhance the competitiveness of U.S. tilapia production.

The tilapia sector of U.S. aquaculture is diverse in geographic location, production systems, and scale of production. Census data show that the volume of tilapia production and the number of farms have contracted since 2015. Clusters of tilapia farms in California, Florida, and North Carolina exhibit striking differences in the economic effects of compliance with their respective regulatory frameworks. Across all tilapia farms, nearly three-fourths of all regulatory filings were at the state or local level. Regulatory costs accounted for 15% of total costs on tilapia farms, with a mean farm cost of $137,611 and a national regulatory compliance cost of $4.4 million per year. The value of time spent by labor and management was a substantial portion of regulatory compliance costs, reducing time available for farming operations, innovation, and improved efficiencies. Smaller tilapia farms experienced greater negative economic effects from regulations. The $32 million in annual lost revenue on U.S. tilapia farms was equivalent to 82% of the total annual sales of tilapia.

This study has exposed the lack of extension assistance for U.S. tilapia producers. Given the high regulatory cost on their farms, additional research and extension assistance is needed to identify more efficient ways to manage farms given the increased stringency of regulations faced by many. Compounding the regulatory costs is the lack of R&D to assess the feasibility of new processing equipment that may have potential for offering a means for U.S. tilapia producers to enter the large seafood fillet market.

The study results show that the regulatory compliance burden is of such an order of magnitude that it likely has contributed to the contraction of U.S. tilapia production and confirms the need for "smarter regulations" that provide necessary oversight without unduly constraining growth. Pathways to increased competitiveness in the U.S. tilapia sector include: (1) regulatory reform to reduce the high compliance burden that could lead to reduced market prices; (2) research and development of cost-effective processing with new tilapia fillet equipment; and (3) marketing research to provide guidance on effective market penetration and growth strategies for the large finfish fillet market in the U.S. Greater research and extension efforts are needed for these pathways to lead to increased competitiveness among U.S. tilapia producers.

This study has reinforced the need for regulatory reform as it relates to aquaculture in the U.S. The regulatory costs on tilapia farms were found to constitute one of the greatest on-farm costs. The disparity in regulatory enforcement and control between developed countries such as the U.S. and EU and developing countries has likely created perverse economic incentives for aquaculture production to shift to countries without the environmental management controls of the U.S. and EU, thereby contributing negatively to environmental quality.

**Author Contributions:** Conceptualization, C.R.E. and J.v.S.; data collection and curation, C.R.E., C.C. and N.B.; analysis, C.R.E.; writing, C.R.E., J.v.S., C.C. and N.B.; editing, C.R.E., J.v.S., C.C. and N.B. All authors have read and agreed to the published version of the manuscript.

**Funding:** This work was supported by the United States Department of Agriculture—National Institute of Food and Agriculture (USDA-NIFA Grant #2018-70007-28827, Accession #1017200).

**Institutional Review Board Statement:** Ethical review and approval were waived for this study because all data were maintained as Confidential Business Data that were coded so that there was no type of identifiable information.

**Data Availability Statement:** The data collected were Confidential Business Data that cannot be shared.

**Acknowledgments:** We thank the tilapia producers across the U.S. who participated in this project and spent the time to provide confidential business information. This study would not have been

possible without their assistance. We also thank the many individuals who assisted with contact lists and who encouraged tilapia farmers to participate in this study. We are grateful for the funding from the United States Department of Agriculture-National Institute of Food and Agriculture (USDA-NIFA Grant #2018-70007-28827, Accession #1017200) as well as the support from the VA Seafood AREC of Virginia Tech University. The findings and conclusions in this publication have not been formally disseminated by the U.S. Department of Agriculture and should not be construed to represent any agency determination or policy. Finally, comments and suggestions by anonymous reviewers contributed to improvements in this paper.

**Conflicts of Interest:** The authors declare no conflict of interest. The funding sponsors were not involved in the conceptualization or design of the study, the analyses or interpretation of the data, or the writing of the manuscript.

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
