# Peer review of "Has the Regulatory Compliance Burden Reduced Competitiveness of the U.S. Tilapia Industry?"

_fishes, doi:10.3390/fishes8030151_

Round 1

Reviewer 1 Report

The topic of the paper is accordance with the topic of the journal. In the basis of the paper, the authors use the methods of descriptive statistics to point out how does regulatory compliance burden impact competitiveness of the U.S. tilapia industry. The paper can be improved with the following:

1. The hypothesis or research question is not clearly emphasized in the paper. Authors should clearly write hypothesis or research question/questions in the section of methodology.

2. Research should be linked with the other similar papers based on topic and/or methodology.

3. Authors used only descriptive statistic analysis. In order to improve research results for example in order to investigate is there significant difference in the total regulatory cost by regulatory category by state or by farm size authors can used chi square or other appropriate statistical tests. These should be in accordance with the hypothesis or research questions.

4. I would recommend removing self-citations that are not  in line with the topic.

5. Figure 3 and figure 4 are not clear. I would recommend presenting answers based on the average score answer.

Author Response

Reviewer 1

The topic of the paper is accordance with the topic of the journal. In the basis of the paper, the authors use the methods of descriptive statistics to point out how does regulatory compliance burden impact competitiveness of the U.S. tilapia industry. The paper can be improved with the following:

  1. The hypothesis or research question is not clearly emphasized in the paper. Authors should clearly write hypothesis or research question/questions in the section of methodology.

RESPONSE: As suggested by this and another reviewer, we have expanded the introduction to more fully and clearly articulate the motivation for this research and the core research questions addressed. Much of this has been added to the introduction, with some mention in the methodology section.

  1. Research should be linked with the other similar papers based on topic and/or methodology.

RESPONSE: In this revision, we have expanded the literature review to more fully link this work to the literature on governance of aquaculture globally and in the U.S. and have worked to improve linkages between this work and other similar papers. In addition, as suggested by the reviewer, we have now added linkages with other similar papers throughout the results section.

  1. Authors used only descriptive statistic analysis. In order to improve research results for example in order to investigate is there significant difference in the total regulatory cost by regulatory category by state or by farm size authors can used chi square or other appropriate statistical tests. These should be in accordance with the hypothesis or research questions.

RESPONSE: The values by state are single values that sum up the total regulatory cost values for each state. Thus, as single values, these cannot be compared statistically. However, the farm size effects, when calculated as a mean value/farm or $/kg per farm are means that we have now compared with an ANOVA, with results presented in Table 11. Similarly, the distributions of the proportions of regulatory costs resulting from each regulatory category have now been compared with a chi-square distribution and results now included in Table 11.

  1. I would recommend removing self-citations that are not in line with the topic.

RESPONSE: We have removed self-citations that are not directly relevant to the research questions addressed in this paper.

  1. Figure 3 and figure 4 are not clear. I would recommend presenting answers based on the average score answer.

RESPONSE: We appreciate the reviewer expressing the concern over the lack of clarity of Figure 3 and Figure 4. However, given how the relevant questions in the survey were framed, an average score would not be appropriate. Respondents were not asked to rank a pre-determined set of responses. Rather, they were asked to report what their top problem was, in their own words. Thus, using stacked bars in those figures shows the overall relative importance of the broad categories of responses provided by respondents, with the colors of the bar segments representing the number of respondents who ranked that category 1, 2, 3, 4, or 5. Given that this convention was used in other, similar farm-level studies that measured the costs of regulatory compliance in aquaculture, we have followed this same convention. We have added clarification to the figure captions on Figures 3 and 4 and have revised the associated text to add clarity for readers.

Reviewer 2 Report

After reading and reviewing the manuscript, the authors used data from a 2021 national survey of U.S. tilapia farmers. In addition, they used a survey tool developed from survey questions on regulatory costs successfully used in previous surveys. The overall goal of this study was to improve understanding of the challenges faced by U.S. tilapia farmers and to assess the extent to which regulatory compliance burdens may contribute to a less competitive U.S. tilapia production.

Finally, after analyzing and discussing the results, they found some specific content, including the maximum regulatory cost, farm regulatory cost, total direct regulatory cost, etc., and put forward a suggestion plan for the U.S. tilapia industry to improve its competitiveness. The authors highlight challenges faced by U.S. tilapia producers and research on the competitiveness of the U.S. tilapia industry.

I found the manuscript interesting, generally well written, however, I have few comments which I describe in detail below:

1. The introduction covers some references, however does not introduce the scientific problem. How will this study contribute beyond the current literature?

2. Lack of experimental design and architecture.

3. Lack of experimental hypothesis.

4. The sample size of some analyzes was too small, which will lead to the validity of the conclusions.

5. Write a separate chapter for the proposed suggestion.

Author Response

Reviewer 2

After reading and reviewing the manuscript, the authors used data from a 2021 national survey of U.S. tilapia farmers. In addition, they used a survey tool developed from survey questions on regulatory costs successfully used in previous surveys. The overall goal of this study was to improve understanding of the challenges faced by U.S. tilapia farmers and to assess the extent to which regulatory compliance burdens may contribute to a less competitive U.S. tilapia production.

Finally, after analyzing and discussing the results, they found some specific content, including the maximum regulatory cost, farm regulatory cost, total direct regulatory cost, etc., and put forward a suggestion plan for the U.S. tilapia industry to improve its competitiveness. The authors highlight challenges faced by U.S. tilapia producers and research on the competitiveness of the U.S. tilapia industry.

I found the manuscript interesting, generally well written,

RESPONSE: Thank you.

However, I have few comments which I describe in detail below:

  1. The introduction covers some references, however does not introduce the scientific problem. How will this study contribute beyond the current literature?

RESPONSE: We have expanded the introduction to more clearly articulate the scientific problem addressed by this study and have stated more explicitly what this study contributes beyond the current literature.

  1. Lack of experimental design and architecture.

RESPONSE: We have now added a more explicit presentation of the study design for this study and added more explicit sub-headings to add clarity on this point. The study design and methodology, however, was not that of an experiment. The study design was that of a descriptive cross-sectional research survey, not an experiment.

  1. Lack of experimental hypothesis.

RESPONSE: We thank the reviewer for pointing out the lack of clarity as to the study design. We have expanded the Methodology and have now added sub-headings to more clearly present the study design, which is that of a descriptive cross-sectional research survey. The study design used was not that of an experiment. While there are various typologies of research, especially for social science research (i.e., descriptive, experimental, correlational, diagnostic, explanatory), this study is a descriptive cross-sectional research survey.

  1. The sample size of some analyzes was too small, which will lead to the validity of the conclusions.

RESPONSE: We understand the concern of the reviewer in that the number of tilapia farms in some states was relatively small, such as the 8 farms identified in California and in North Carolina. Nevertheless, the response rates (which is the sample size) in these states were 87.5% and 62.5%, respectively, indicating that our dataset included very high percentages of all the farms in those states. We have added a paragraph on the limitations of the study, which is that, since tilapia farms are scattered across the U.S., the total numbers in some states are relatively small. Nevertheless, our data do capture the majority of the tilapia production in those states.

  1. Write a separate chapter for the proposed suggestion.

RESPONSE: As suggested, we have added a short section following the Discussion section on Potential Pathways for Growth of U.S. Tilapia Production that summarizes suggestions and recommendations.

Reviewer 3 Report

The present manuscript is clearly conceived, well structured and clearly written. It depicts the entity of several cost concerning compliance with current regulations on finfish aquaculture production in a number of states within the USA.

In the manuscript, authors implicitly assume that Tilapia production costs are due to regulatory duties, and those duties are responsible of some throubles faced by Tilapia producers. However, other factors not considered here could influence the dynamics of the Tilapia market as, e.g., substitution of Tilapia by other fish products as a result of deeper concern of US public on the impact of Tilapia farming, loss of spending capacity by the public, or competition at national level with other fish species only recently reared. Authors do not provide information in support or disregard of these and other potential possibilities.

The present MS puts heavy emphasis on different types of cost directly and indirectly related to compliance with state-level regulations. Most of those regulations seem related to environmental, ecological, and sanitary concerns. Authors claim that the costs driven by state regulations are excessively high and that smartest [I intend less costly to comply] regulations are needed. They deserve a large proportion of the manuscript to illustrate in detail the magnitude of the present regulation compliance costs, but do not provide any information about what changes could made the regulation framework smarter. Such definition is important but lacking from the present manuscript.

Generally speaking, the amount of environmental, ecological and health regulations are directly related with the societal concerns on these matters, which in turn correlates with the degree of social development and wellness. Hence smarter regulations would be possibly e.g. harmonized (between states), more clearly stated, and less time-consuming than current ones. However, you should not expect less stringent regulations from developed and empowered societies. It is probably a better strategy for enduring in the bussiness to increase revenues from a more elaborated product (e.g. fillets) that can reach a larger market, while stressing differences in taste and ecological footprint with respect to imported alternatives.

In my opinion the MS is too heavily focused in the cost of Tilapia production compared with other reared species in the USA. Such comparison is of limited utility since the technologies, markets and processes are different on depending of the focused species. What is needed is comparing regulation cost among countries of equal or similar social systems and level of technological development were Tilapia (or another similar species from the economic point of view) is grew for selling it alive. If such conditions cannot be matched, then compare regulation burden and compliance cost for live Tilapia in the US with other production of whatever other species of reared fish in similarly developed countries. This is particularly feasible since the cost of keeping tilapia alive is charged to the whole-sealer or distributor and it is therefore not needed to correct the total cost of production in order to account for this particular way to offer the product. If national rearing conditions are roughly comparable, then look for the national regulation systems and identify those elements making some national regulation systems smarter than others. Finally, try to translate the identified elements into the US regulation context and provide specific recommendations to make the current regulations better that they are actually.

Of course such comparison should be made among analogous costs in countries with similar social and economic systems and level of development. Otherwise, apply weighted correction factors. In a certain way, it is about to check for some reference (control) situations to which comapre your focused case.

Another important point which deserves further development is the identification of the factors that would enable Tilapia producers to access the fillet market. The current live Tilapia production could go into trouble simply because too restricted to strive in the ever-changing recent market conditions.

Some minor comments (mostly edits) are provided in the PDF herewith provided.

Author Response

Reviewer 3

The present manuscript is clearly conceived, well structured and clearly written.

RESPONSE: Thank you.

It depicts the entity of several cost concerning compliance with current regulations on finfish aquaculture production in a number of states within the USA.

In the manuscript, authors implicitly assume that Tilapia production costs are due to regulatory duties, and those duties are responsible of some troubles faced by Tilapia producers. However, other factors not considered here could influence the dynamics of the Tilapia market as, e.g., substitution of Tilapia by other fish products as a result of deeper concern of US public on the impact of Tilapia farming, loss of spending capacity by the public, or competition at national level with other fish species only recently reared. Authors do not provide information in support or disregard of these and other potential possibilities.

RESPONSE: The reviewer’s point is well taken that we had not devoted much space in the paper to these other potential causes of the contraction in the U.S. tilapia industry. As per the reviewer’s suggestion, we have now expanded the discussion section of the paper to discuss these possibilities.

The present MS puts heavy emphasis on different types of cost directly and indirectly related to compliance with state-level regulations. Most of those regulations seem related to environmental, ecological, and sanitary concerns. Authors claim that the costs driven by state regulations are excessively high and that smartest [I intend less costly to comply] regulations are needed. They deserve a large proportion of the manuscript to illustrate in detail the magnitude of the present regulation compliance costs, but do not provide any information about what changes could made the regulation framework smarter. Such definition is important but lacking from the present manuscript.

RESPONSE: We appreciate the reviewer’s point and, as suggested, have added a new section following the Discussion section of the paper that we have entitled “Potential Pathways for Growth of U.S. Tilapia Production” in which we discuss potential options for a “smarter” regulatory framework and some other potential options. We believe that this section has improved the overall paper and thank the reviewer for this suggestion.

Generally speaking, the amount of environmental, ecological and health regulations are directly related with the societal concerns on these matters, which in turn correlates with the degree of social development and wellness. Hence smarter regulations would be possibly e.g. harmonized (between states), more clearly stated, and less time-consuming than current ones. However, you should not expect less stringent regulations from developed and empowered societies.

RESPONSE: This is true, and most farmers interviewed appreciated the benefits of the well-ordered society and environmental and work place benefits of stringent regulations. The farmers interviewed did not advocate for less stringent regulations, but less redundant and more streamlined and efficient monitoring and reporting systems. We thank the reviewer for this comment and have worked to make this point more clearly in the presentation of both the results, discussion, and the new section on Possible Pathways for Growth of U.S. Tilapia Production.

It is probably a better strategy for enduring in the business to increase revenues from a more elaborated product (e.g. fillets) that can reach a larger market, while stressing differences in taste and ecological footprint with respect to imported alternatives.

RESPONSE: What the reviewer suggests is the Holy Grail for U.S. tilapia producers – to be able to compete in the much larger finfish fillet market in the U.S. Until now, however, the assumption has been that the relatively lower dressout yield of tilapia and the overall higher costs of labor in the U.S. would mean that U.S.-produced tilapia fillets would not be competitive with imports. However, there have been some new technological developments that may change that situation. We had very briefly mentioned those in the original version of the paper, but have now elaborated more on this point in the new section on Possible Pathways for Growth of U.S. Tilapia Production.

In my opinion the MS is too heavily focused in the cost of Tilapia production compared with other reared species in the USA. Such comparison is of limited utility since the technologies, markets and processes are different on depending of the focused species. What is needed is comparing regulation cost among countries of equal or similar social systems and level of technological development were Tilapia (or another similar species from the economic point of view) is grew for selling it alive. If such conditions cannot be matched, then compare regulation burden and compliance cost for live Tilapia in the US with other production of whatever other species of reared fish in similarly developed countries. This is particularly feasible since the cost of keeping tilapia alive is charged to the whole-sealer or distributor and it is therefore not needed to correct the total cost of production in order to account for this particular way to offer the product. If national rearing conditions are roughly comparable, then look for the national regulation systems and identify those elements making some national regulation systems smarter than others. Finally, try to translate the identified elements into the US regulation context and provide specific recommendations to make the current regulations better that they are actually. Of course, such comparison should be made among analogous costs in countries with similar social and economic systems and level of development. Otherwise, apply weighted correction factors. In a certain way, it is about to check for some reference (control) situations to which compare your focused case.

RESPONSE: The study proposed above by the reviewer would be very interesting indeed. Unfortunately, we do not know of any similar studies in other countries that have measured the detailed farm-level regulatory compliance cost burden of tilapia or any other aquaculture species. There have been some studies of regulatory burdens in the EU, but those used national and international data, and did not measure actual farm-level regulatory costs. Thus, we are not aware of an adequate dataset to make the suggested comparison. We are intrigued by the thought, however, and have mentioned such additional work as a recommendation for additional research.

Another important point which deserves further development is the identification of the factors that would enable Tilapia producers to access the fillet market. The current live Tilapia production could go into trouble simply because too restricted to strive in the ever-changing recent market conditions.

RESPONSE: What the reviewer suggests is the Holy Grail for U.S. tilapia producers – to be able to compete in the much larger finfish fillet market in the U.S. Until now, however, the assumption has been that the relatively lower dressout yield of tilapia and the overall higher costs of labor in the U.S. would mean that U.S.-produced tilapia fillets would not be competitive with imports. However, there have been some new technological developments that may change that situation. We had very briefly mentioned those in the original version of the paper, but have now elaborated more on this point in the new section on Potential Pathways for Growth of U.S. Tilapia Production.

Some minor comments (mostly edits) are provided in the PDF herewith provided.

RESPONSE: We have gone through these edits carefully and made the changes necessary.

Reviewer 4 Report

THE WORK IS VERY IMPORTANT FOR THE US  TILAPIA SECTOR (AND NOT ONLY) SINCE IT IS COMING UP TO THE SURFACE THE ADMINISTRATIONAL COST OF THE INDUSTRY. IT IS A PIONEER EFFORT TO QUANTIFY THE LOSSES FROM THE INSTITUTIONAL RISKS. THE ,  IS A GOOD BENCHMARKING MODEL FROM OTHER INDUSTRIES.

THE ONLY COMMENT THAT I HAVE IS ON THE PAGE FORMATION AND THE SPLIT OF THE TABLES AND LEGENDS IN THE TEXT THAT HAS TO BE AVOIDED.

Author Response

Reviewer 4

THE WORK IS VERY IMPORTANT FOR THE US  TILAPIA SECTOR (AND NOT ONLY) SINCE IT IS COMING UP TO THE SURFACE THE ADMINISTRATIONAL COST OF THE INDUSTRY. IT IS A PIONEER EFFORT TO QUANTIFY THE LOSSES FROM THE INSTITUTIONAL RISKS. THE ,  IS A GOOD BENCHMARKING MODEL FROM OTHER INDUSTRIES.

RESPONSE: Thank you.

THE ONLY COMMENT THAT I HAVE IS ON THE PAGE FORMATION AND THE SPLIT OF THE TABLES AND LEGENDS IN THE TEXT THAT HAS TO BE AVOIDED.

RESPONSE: We agree. Hopefully, that will be worked out in the layout process.